# Non-Native Turtles (Chelydridae) in Freshwater Ecosystems in Italy: A Threat to Biodiversity and Human Health?

**DOI:** 10.3390/ani12162057

**Published:** 2022-08-12

**Authors:** Giuseppe Esposito, Luciano Di Tizio, Marino Prearo, Alessandro Dondo, Carlo Ercolini, Gianpiero Nieddu, Angelo Ferrari, Paolo Pastorino

**Affiliations:** 1The Veterinary Medical Research Institute for Piedmont, Liguria and Valle D’Aosta, Via Bologna 148, 10154 Turin, Italy; 2Regional Reference Centre for Biodiversity of Aquatic Environments, Via Maritano Lino 22, 10051 Avigliana, Italy; 3*Societas Herpetologica Italica*, Personal Address: Via Federico Salomone 112, 66100 Chieti, Italy; 4Freelance Veterinarian, Via Alcide De Gasperi 15, 27100 Pavia, Italy

**Keywords:** alien species, biological invasion, freshwater, reptiles, testudines, zoonoses

## Abstract

**Simple Summary:**

Marketed globally, freshwater turtles are popular pets. However, the introduction of non-native species can pose a serious threat to biodiversity as well as to human health as carriers of potentially zoonotic pathogens. This update reports the distribution of two species of the Chelydridae family, i.e., the snapping turtle (*Chelydra serpentina*) and the alligator snapping turtle (*Macrochelys temminckii*). Their potential impact on human health and biodiversity is also discussed.

**Abstract:**

Marketed globally, freshwater turtles are popular pets. Two species of the Chelydridae family are increasingly reported in Italy: the snapping turtle (*Chelydra serpentina*) and the alligator snapping turtle (*Macrochelys temminckii*). Both pose potential threats to public safety and habitat biodiversity. This update reports on their distribution and impact on biodiversity and human health. The recent increase in the number of *C. serpentina* in urban and rural areas suggests illegal importation into the country. Findings are reported for the north (35% and 100% for *C. serpentina* and *M. temminckii*, respectively) and the central-northern regions (60% for *C. serpentina*), predominantly Umbria and Latium, and the Tiber River catchment area in particular. Because omnivorous, Chelydridae species can affect native biodiversity; because they are carriers of pathogens, they endanger public health. Monitoring plans need to take account of this neglected threat.

## 1. Introduction

Globalization, coupled with expanding trade and transport, has opened pathways to biological invasion [1]. The steady accumulation of invasive alien species (IAS) [2] is projected to impact on socio-ecological conditions globally. Driven by transportation, climate change, and socio-economic change [3], the movement of biological species is closely related to the environment [4]. In general, the share of reported species by environment has decreased between 2012 and 2019: freshwater (44%), marine (43%), terrestrial (10%), and transitional (3%), with most (42%) reported for Europe [4].

The total cost of IAS in Europe between 1960 and 2020 was estimated at EUR 116 billion, albeit considerably underestimated [5]. Introduced by human activity (deliberately or inadvertently), the arrival of IAS has serious repercussions on the stability of ecosystems and human life. In Italy alone, more than 3000 alien species are reported, with an overall cost (between 1990 and 2020) estimated at about EUR 705 million [6]. Under European Union (EU) Regulation 1143/2014 [7] and Italian Legislative Decree no. 230/2017 [8], bans and measures have been set to prevent and manage the introduction and spread of IAS and mitigate their negative effect on the environment and biodiversity.

Marketed globally, freshwater turtles are popular pets, especially with children. Several non-native turtle species have been reported in recent years in European terrestrial and aquatic natural environments [9]. They are generally opportunistic species, colonising various freshwater habitats generally close to urban areas. Reports in urban areas appear to be increasing both in the number of individuals and in the variety of species, including those whose possession is prohibited. Among them, for example, species such as the pond slider (*Trachemys scripta*), the river cooter (*Pseudemys concinna*), and the false map turtle (*Graptemys pseudogeographica*) have been introduced for nursery purposes and as pets [9].

When intentionally abandoned due to rapid growth or by accidental escape, however, they enter wild or semi-natural habitats. The most widely sold is the American pond slider (*Trachemys scripta*). It was listed among the IAS of EU concern in 2016, and ranked among the top 100 most harmful IAS [7]. Reports of species of the Chelydridae family (Gray, 1831) have become increasingly frequent in Europe [9]. They are mainly carnivorous and tendentially nocturnal species. Moreover, they are potentially dangerous for public health and safety and their sale and possession is illegal in Italy [8].

Within the Chelydridae family (Gray, 1831), widespread in the Americas, are two genera *Chelydra* and *Macrochelys*, which include species of the largest freshwater turtles living today. The snapping turtle (*C. serpentina*) (Linnaeus, 1758) has been considered the only species of the genus *Chelydra* that is found from southern Canada to the southern United States, and the one most frequently exported [9]. They grow to considerable size: the males (larger) can reach 50 cm in carapace length and about 20 kg in live weight [9].

The alligator snapping turtle (herein referred to as *M. temminckii*) (Troost in Harlan, 1835) is the largest freshwater turtle among the species found in North America; its carapace can reach up to 66 cm in length and its body weight up to hundreds of kilos [9]. Its native range is limited to the river systems that flow into the northern Gulf of Mexico from Texas to Florida [10,11]. Recent taxonomic revisions have been proposed based on morphological and genetic data, with the identification of new species and subspecies [12,13]. Our study does not consider these revisions because we believe it impossible to distinguish the species solely by morphological characteristics and establish the area of origin of the turtles imported into Italy.

In Italy, alligator and snapping turtles were sold via the pet market until banned [14,15,16]. They were placed on the list of dangerous animals, together with the Caspian turtle (*Mauremys capsica*) (Gmelin, 1774).

According to the International Union for Conservation of Nature (IUCN), IAS are a leading cause of the decline in biodiversity worldwide. IAS threaten the existence of many native species and ecosystems and hasten the destruction of habitats. Assessments of their impact on local ecosystems and targeted eradication and/or containment plans are needed. The rise in the number of reports and/or captures of adult individuals of *C. serpentina* and *M. temminckii* is cause for alarm for alarm in Italy. Accordingly, we analysed the current distribution for non-native chelydrid turtles and characterized their potential threat to biodiversity and human health.

## 2. Materials and Methods

### 2.1. Update of Chelydridae in Italy: Literature Search

We carried out a search of print and online literature to find recent information about the two species. To achieve this, we consulted national and regional atlases of reptiles, scientific publications, and mass media articles with national or local circulation. The electronic databases Google Scholar (https://scholar.google.it/; accessed on 23 June 2022) and Scopus (https://www.scopus.com/; accessed on 26 July 2022) were queried using the search terms “*Chelydra serpentina*” and “*Macrochelys temminckii*” combined with “Italy, distribution, Mediterranean distribution”. Very few of the retrieved records referred to reports for Italy.

### 2.2. Study Area

Italy is a central-southern European country (Figure 1) rich in mountains and streams. The major rivers arise in the Alps, while the rivers with a short course and a torrential regime arise in the Apennines. The Po is longest river (about 650 km long, average flow rate 1460 m^3^/s, basin roughly 70,000 km^2^); it drains the Po Valley from west to east before emptying into the Adriatic Sea. The three largest pre-Alpine lakes are Garda, Maggiore, and Como, which is also the deepest (about −410 m).

Italy is located near the centre of the temperate zone of the northern hemisphere; its Mediterranean climate is strongly influenced by the seas surrounding the peninsula. According to Köppen classification [17], there are three climatic zones: boreal-polar in the north, temperate in most areas, and arid in the south, including the islands of Sicily and Sardinia.

## 3. Results and Discussion

Between 2000 and 2021, 40 reports of the snapping turtle (*Chelydra serpentina*) were made; 39 (97.5%) of the 41 specimens identified were isolated individuals (sighted and/or captured); 2 (2.5%) were found in 2020, 1 of which in Lake Lucinasco (Figure 2). Only 4 (9.8%) individuals were identified up to 2010, and 37 (90.2%) have been reported since 2011. No individuals have been reported for Sicily or Sardinia (Figure 1); most reports come from the north and the central-northern areas of the country and one report for the Adriatic coast in 2002 [18]. Since 2011, reports have come from the regions of Umbria and Latium (*n* = 14; 34.1%) and the Tiber River basin in particular [19,20,21] (Table 1 and Table 2).

Between 2012 and 2021, there were only five reports of the alligator snapping turtle (*Macrochelys temminckii*) (Figure 1). While the incidence does not appear significant in absolute terms, it raises cause for concern since four of these five reports come from the same site, Lake Lucinasco (Imperia, Liguria): in addition to the one individual found in 2020 and the three in 2021 (Figure 2), two individuals of *C. serpentina* were reported in 2020 at the same site. However, the turtles were captured and removed from the natural environment.

The difference in the number of finds between the two species probably stems from their behaviour. The snapping turtle (*C. serpentina*) is usually active during the day and only sometimes at night [22]. After coming out of hibernation in late April, the turtle’s activity varies between males and females throughout the reproductive season (May–August) [23]. Differently, the alligator snapping turtle (*M. temminckii*) is predominantly nocturnal [24,25]; juveniles follow a different activity pattern (Spangler, 2017) [25], thus reducing the risk of nocturnal predation until they have grown in size [26]. What little is known about its ecology is that it is a bottom species preferring deep waters in late summer and midwinter [27,28], which explains the low number of sightings. Many of the individuals of both species were adults that very probably were abandoned because of difficult management once grown.

In America, *C. serpentina* is not classified as an endangered species, whereas *M. temminckii* is listed as a vulnerable species by the World Union for Conservation of Nature (IUCN); nonetheless, both are included in the Appendix III of the Washington Convention (CITES, Convention on International Trade in Endangered Species). The illegal possession and trade of these animals are punishable by Law no. 150/1992 [14]. Additionally, worth noting, the two recognized species of *Macrochelys* are proposed for listing as “threatened” in the U.S. Endangered Species Act [29,30].

Both species are omnivorous and can negatively affect the aquatic communities of colonized environments through the predation of a variety of animal species [31,32]. For instance, the analysis of the gastrointestinal tract contents of *M. temminckii* revealed their ability to prey on a wide range of organisms (e.g., fish, shrimp, molluscs, turtles, insects, and small mammals) [32]. However, vegetation could be ingested incidental to capture of animal prey [32]. Furthermore, they can compete with the European pond turtle (*Emys orbicularis*) (Linnaeus, 1758) [33] for food, spawning sites, basking sites, etc. or prey on it. The pond turtle is a protected species in most countries where present; it is classified as vulnerable in Europe or as endangered in several European countries including Italy.

In addition to their negative impact on biodiversity, *C. serpentina* and *M. temminckii* pose a potential health threat to humans. They primarily pose a significant zoonotic risk to pet owners, zookeepers, and veterinarians [34]. Immunocompromised people, such as children, pregnant women, chronic disease patients, and those on immunosuppressive therapy, are particularly vulnerable [35]. Direct contact with turtles or contaminated environments, such as soil and water, can spread potential pathogenic microorganisms such as bacteria or viruses. For example, several bacteria infect the gastrointestinal tract of turtles and contaminate the environment through shedding. Neglecting to clean the pond/aquaria increases the bacterial load in the water environment, potentially causing diseases in turtles. Humans can also become infected with pathogenic bacteria through direct or indirect contact with turtles. If pet owners do not maintain proper hygiene (i.e., washing hands or using gloves) after touching their turtles or water, the bacteria can cause human infections. Humans are primarily infected through the fecal–oral route, but infection can also be transmitted through bites or claw scratches [36]. Contamination from turtles can cause severe diarrhea, septicemia, gastrointestinal disorder, sepsis, and in severe cases, death [37]. As a result, some countries set up some rules and regulations to address this issue. For example, small turtles were prohibited from sale and distribution in the USA and Canada to prevent the horizontal spread of pathogenic bacteria from turtles to humans [38]. This ban was a strong public health measure aimed at preventing the spread of zoonotic bacteria [39]. Even though pet turtles are widely raised in many developed and developing countries, such rules do not exist [38]. In the literature, most studies focused the attention on *Salmonella* spp. found in pet turtles. For example, *Salmonella hartford* from a pet turtle was isolated from a 7-month-old patient in 1963, and the patient’s family owned a pet turtle [40]. In 1964 and 1965, a total of 100 cases of *S. java* and *S. panama* infections were reported in pet turtles [41,42]. In the late 1970s, the infection rate of *Salmonella* spp. in pet turtles from United States was 85%, and turtles were responsible for about 14% of nontyphoidal salmonellosis [43]. Many reptiles, including testudines, are carriers of salmonellosis in humans [44,45]. In 1975, the U.S. Food and Drug Administration banned the sale and import of pond turtles because of the thousands of cases of salmonellosis it causes every year [46,47]. Contamination by *Salmonella* spp. of clinical interest by turtles is very rarely reported for Europe [48,49]. Other human pathogenic bacteria isolated in freshwater turtle include *Aeromonas* spp., *Citrobacter* spp., *Enterobacter* spp., *Klebsiella* spp., *Proteus* spp., and *Serratia* spp. [50].

Wild and domestic animals act as vectors to leptospirosis for humans, a systemic disease that is the most widespread zoonosis worldwide [51]. Transmission to humans commonly occurs through contact with the urine of animal reservoirs (e.g., rodents and livestock animals), either directly or via contaminated surfaces [51,52]. In this regard, herpetofauna, including reptiles and amphibians, turn out to be carriers of different serotypes of *Leptospira* spp. [53]. In recent years, there have been increasing reports of this zoonotic bacterium in some freshwater turtle species, especially from urban environment. For instance, occurrence has been highlighted in the American pond slider (*Trachemys scripta* spp.) captured from urban ponds of the metropolitan city of Turin, Italy [53], and in a natural park located in Valbrembo of Northern Italy [54], in Geoffroy’s side-necked turtle (*Phrynops geoffroanus*) from urban stream in the city of Jaboticabal, Brazil [55], and in Blanding’s turtles (*Emydoidea blandingii*) from an urban setting DuPage County, IL, USA [56]. Furthermore, a recent study in Italy on specimens found dead after the hibernation period showed that these reptiles can carry pathogens such as Chlamydiaceae [57].

## 4. Conclusions

The recent increase in the distribution of freshwater turtles of the Chelydridae family, especially in central Italy, raises cause for alarm. They endanger human health and safety and threaten many native species and ecosystems. Known for their bite and aggressiveness when out of water and if annoyed, a bite can result in injury or amputation of fingers. They are often abandoned in public and/or rural areas (e.g., parks, rivers, and canals) mainly because difficult to manage when adults. Urban parks are green areas usually located within a city, with the purpose of providing a recreational space in contact with nature for citizens and other visitors. These areas are often inhabited by freshwater turtles, probably due to accidental escapes or via voluntary releases. So, even if they are seemingly harmless, reptiles may serve as both reservoirs and accidental hosts for various pathogens to humans. Monitoring plans are needed to keep the expansion of these reptiles under control and to evaluate their reproductive capacity in the perspective of climate change.

## Figures and Tables

**Figure 1 animals-12-02057-f001:**
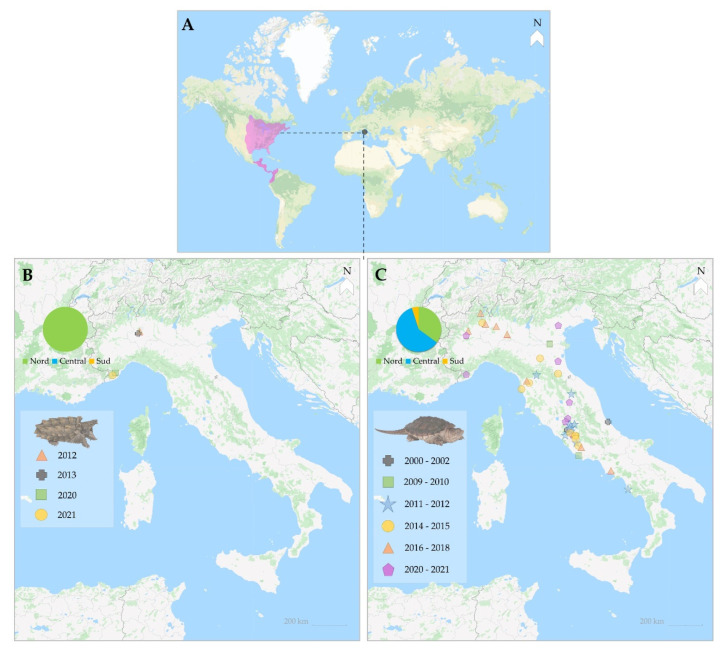
Native distribution (in fuchsia (**A**)) and distribution of Chelydridae in aquatic environments in Italy from 2000 to 2021 ((**B**), alligator snapping turtle (*Macrochelys temminckii*); (**C**), snapping turtle (*Chelydra serpentina*)).

**Figure 2 animals-12-02057-f002:**
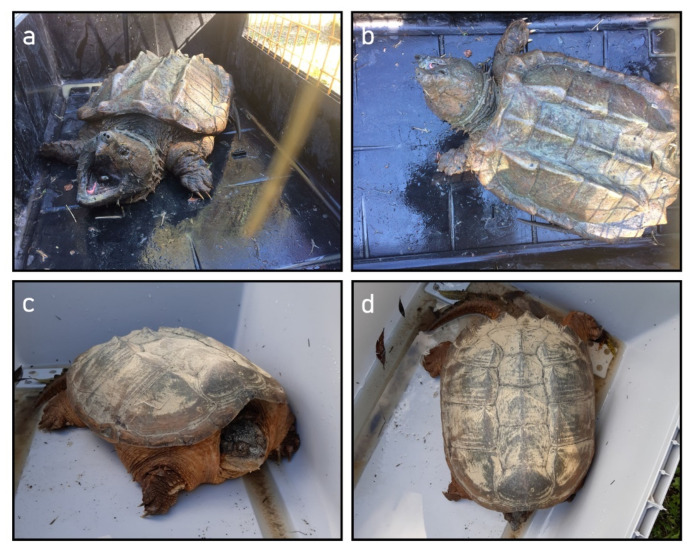
Adult individuals of the alligator snapping turtle (*Macrochelys temminckii*) (**a**,**b**) and the snapping turtle (*Chelydra serpentina*) (**c**,**d**) from Lake Lucinasco (Liguria, Italy): (**a**,**c**) front view and head detail; (**b**,**d**) top view with detail of the carapace. Photo courtesy of Vasco Menconi (**a**,**b**), and Manuel Maggioli (**c**,**d**).

**Table 1 animals-12-02057-t001:** Alligator snapping turtle (*Macrochelys temminckii*) reported in aquatic environments in Italy.

Site	Region	Discovery Area	Year	TW ^#^	TCL ^#^	n°	References
Binasco/Milano	Lombardia	Rice fields	2012	–	–	1	[58]
Milano	Lombardia	Urban area	2013	15	100 *	1	[59]
Lucinasco	Liguria	Artificial pond	2020	–	–	1	[60]
Lucinasco	Liguria	Artificial pond	2021	–	–	3	[61]

* Whole individual including tail. ^#^ TW = total weight, TCL = total carapace length; expressed in kilograms (kg) and centimetres (cm), respectively.

**Table 2 animals-12-02057-t002:** Snapping turtle (*Chelydra serpentina*) reported in aquatic environments in Italy.

Site	Region	Discovery Area	Year	TW ^#^	TCL ^#^	n°	References
Nepi	Lazio	Stream	2000	–	–	1	[19]
Pescara	Abruzzo	Urban area	2002	–	–	1	[18]
Canaro	Veneto	River	2009	10	–	1	[62]
Parco Nazionale del Circeo	Lazio	Rural area	2010	20	70	1	[63]
Anguillara	Lazio	Rural area	2011	20	60	1	[64]
Noce Alta	Campania	Rural area	2011	5	–	1	[65]
Gallese	Lazio	River	2011	–	–	1	[20]
Filacciano	Lazio	River	2011	–	–	1	[20]
Botaccione	Umbria	River	2012	–	–	1	[66]
Badia a Pacciana	Toscana	Rural area	2012	–	–	1	[67]
Stimigliano	Lazio	River	2012	–	–	1	[68]
Fidene	Lazio	Urban area	2012	–	–	1	[69]
Ponzano Romano	Lazio	River	2012	–	–	1	[20]
Poggio Mirteto	Lazio	River	2012	–	–	1	[20]
Cusercoli	Emilia-Romagna	River	2014	24	~100	1	[70]
Livorno	Toscana	Urban area	2014	2.8	33	1	[71]
Casinalbo	Emilia-Romagna	Urban area	2014	>5	30	1	[72]
Castelnuovo di Porto	Lazio	Urban area	2014	–	–	1	[20]
Monterotondo	Lazio	River	2014	–	–	1	[20]
Monterotondo	Lazio	River	2014	–	–	1	[20]
Roma	Lazio	River	2014	–	–	1	[20]
Oleggio (fraz. San Giovanni)	Piemonte	Rural area	2015	–	25	1	[73]
Velletri	Lazio	Rural area	2015	–	–	1	[74]
Fucecchio	Toscana	River	2015	–	–	1	[75]
Strambino (fraz. Crotte)	Piemonte	Rural area	2016	–	–	1	[76]
Cori	Lazio	Rural area	2016	7	–	1	[77]
Milano	Lombardia	River	2016	–	–	1	[78]
Pomigliano d’Arco	Campania	Urban area	2018	–	–	1	[79]
Castiglione d’Adda	Lombardia	Rural area	2018	8	35	1	[80]
Fucecchio	Toscana	Rural area	2018	15	–	1	[81]
Arconate	Lombardia	Urban area	2018	–	40	1	[82]
Parco Villa Pallavicino	Piemonte	Pond	2018	>6	–	1	[83]
Monterotondo	Lazio	River	2020	–	–	1	[84]
Lucinasco	Liguria	Artificial pond	2020	–	–	2	[85]
Codrignano	Emilia-Romagna	Artificial pond	2020	–	–	1	[86]
Casier	Veneto	River	2021	~10	–	1	[87]
Collegno	Piemonte	River	2021	–	–	1	[88]
Cannara	Umbria	Urban area	2021	–	25	1	[89]
Narni	Umbria	Artificial pond	2021	–	–	1	[90]
Otricoli	Umbria	River	–	–	–	1	[89]

^#^ TW = total weight, TCL = total carapace length; expressed in kilograms (kg) and centimetres (cm), respectively.

## Data Availability

Not applicable.

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
