# Peer review of "Non-Native Turtles (Chelydridae) in Freshwater Ecosystems in Italy: A Threat to Biodiversity and Human Health?"

_animals, 2022, doi:10.3390/ani12162057_

Round 1

Reviewer 1 Report

I have read the manuscript “Non−native turtles (Chelydridae) in aquatic ecosystems in Italy:  a threat to biodiversity and human health?” by Giuseppe Esposito and colleagues, submitted to the Animal magazine. I think, it is interesting, and well written, communications on two alien species of turtles, i.e. the snapping turtle (Chelydra serpentina) and the alligator snapping turtle (Macrochelys temminckii), in Italy. Such data on presence of non-native species are valuable, in this case especially as the turtles could have various effects on ecosystem and human health.
I have some remarks to this version of the manuscript –
I hope that they can be used to improve it.

In the present version, the aim of the study is not adequate stand out. I think that the last paragraph of the introduction section could be improved.

Figure 1. The figures A and B are not well legible. Additionally, I am not sure if the term “and translocation” is the best one for the legend – it was rather no translocation program.
It would be better to sign the part of the figure as “A”, “B” and “C” (now, the first part is without any sign).

Table 1. The table could be improved, I think. First, it would be better to divide the table for two ones (or two clear parts), and remove the column “Species”. In the present form, the table is organized by the species, and then organized by the date. It could be confusing for readers, I think.
The word “Total” is not needed, or should be situated in the different part of the table/line.

Specific comments:
Consider to change the word “specimen” to “individual”. I think, the word “individual” would be better, for example see lines 73-74 “live specimens of mammals and reptiles”.

line 69 – on the end of the line should be full stop, not semicolon, I think,

line 74: unnecessary bracket ‘]’, or lack of citation,

line 98: km2 – “2” should be written as superscript,

List of references should be corrected, for example scientific names of some species should be written in italics (e.g., No. 27 Riedle et al. 2006, and No. 32 Cadi and Joly 2003).

Author Response

Response to #Reviewer 1

  • I have read the manuscript “Non−native turtles (Chelydridae) in aquatic ecosystems in Italy:  a threat to biodiversity and human health?” by Giuseppe Espositoand colleagues, submitted to the Animal magazine. I think, it is interesting, and well written, communications on two alien species of turtles, i.e. the snapping turtle (Chelydra serpentina) and the alligator snapping turtle (Macrochelys temminckii), in Italy. Such data on presence of non-native species are valuable, in this case especially as the turtles could have various effects on ecosystem and human health. I have some remarks to this version of the manuscript – I hope that they can be used to improve it. In the present version, the aim of the study is not adequate stand out. I think that the last paragraph of the introduction section could be improved.

Dear Reviewer, thank you very much for your suggestions which will certainly improve this manuscript. We share the fact that such data (often overlooked) are important both from an environmental and human health perspective.

  • Figure 1. The figures A and B are not well legible. Additionally, I am not sure if the term “and translocation” is the best one for the legend – it was rather no translocation program.
    It would be better to sign the part of the figure as “A”, “B” and “C” (now, the first part is without any sign).

Thanks for the suggestion. The map was changed and replaced the word “translocation” with “distribution”. Regarding the image resolution, it could be a problem when uploading the file as the initial resolution is >300 dpi.

  • Table 1. The table could be improved, I think. First, it would be better to divide the table for two ones (or two clear parts), and remove the column “Species”. In the present form, the table is organized by the species, and then organized by the date. It could be confusing for readers, I think. The word “Total” is not needed, or should be situated in the different part of the table/line.

Thanks for the suggestion. Done.

Specific comments:

  • Consider to change the word “specimen” to “individual”. I think, the word “individual” would be better, for example see lines 73-74 “live specimens of mammals and reptiles”.

Thanks for the suggestion. Done.

  • line 69 – on the end of the line should be full stop, not semicolon, I think.

Thanks for the suggestion. Done.

  • line 74: unnecessary bracket ‘]’, or lack of citation.

Thanks you, done.

  • line 98: km2 – “2” should be written as superscript.

Thank you for your observation, done.

  • List of references should be corrected, for example scientific names of some species should be written in italics (e.g., No. 27 Riedle et al. 2006, and No. 32 Cadi and Joly 2003).

Thank you for your observation, done.

Reviewer 2 Report

See attached comments and suggestions. There are many long and awkward sentences, especially in the Introduction, owing to the use of semicolons. Suggest simplifying the sentence structure into concise statements and avoid using too many semicolons in the manuscript.

Author Response

Response to #Reviewer 2

  • See attached comments and suggestions. There are many long and awkward sentences, especially in the Introduction, owing to the use of semicolons. Suggest simplifying the sentence structure into concise statements and avoid using too many semicolons in the manuscript.

Thanks for the suggestion. Done.

  • More specifically, the referenced 2014 regulation [#7] identified as Trachemys scripta elegans (Wied-Neuwied, 1838), or red-eared slider. The subsequent regulation in 2016 identified as simply Trachemys scripta (Schoepff, 1792). https://eur-lex.europa.eu/legal-content/EN/TXT/PDF/?uri=CELEX:32016R1141&from=EN. Suggest using the latter as the reference.

Thanks for the suggestion, done. The reference link has been changed, and the words “and subsequent amendments” have been added to the reference to Regulation (EU) No 1143. (i.e., current consolidated version: 14/12/2019).

  • Insert “Gray (1831)” after family as in following paragraph. Where are the reports more frequent, Europe in general?

Thanks for the suggestion, done.

  • Simplify the sentence for lines 45-47, "Two introduced species of chelydrid turtles are potentially dangerous for public health and safety and their sale and possession is illegal in Italy." Reference for this statement?

Thanks for the suggestion, done.

  • This is a confusing description of the genus Chelydra. The subspecies of C. serpentina are no longer recognized (see https://ssarherps.org/wp-content/uploads/2017/10/8th-Ed-2017-Scientific-and-Standard-English-Names.pdf) so there is no reason to refer to them as such in the manuscript.

Instead, suggest the authors note there are 3 species of Chelydra (serpentina, rossignonii, and acutirostris in North, Central, and South America, respectively) and the North American species C. serpentina is the most frequently exported (Reference?).

Thanks for the suggestion, done.

  • Since it is later stated that different species of Macrochelys have been identified, the authors need to indicate that they are referring to the multiple species as "M. temminckii" in their manuscript. Perhaps change the first sentence to:

The alligator snapping turtle (herein referred to as M. temminckii)...

Thanks for the suggestion, done.

  • Suggest "native range" rather than "original diffusion in nature".

Thanks for the suggestion, done.

  • Should also include citation for Folt and Guyer (2015, Zootaxa 3947: 447–450) since this is the currently accepted delineation of Macrochelys species in the U.S.

Thanks for the suggestion, done. Please, see reference number 12.

  • Lines 69-74, this is an awkward complex sentence. Split into 2-3 separate sentences to make your point. It also appears this information is repeated in the subsequent discussion.

Thanks for the suggestion, done.

  • ...cause for alarm in Italy. Accordingly, we analyzed the current distribution for non-native chelydrid turtles and characterized their potential threat to biodiversity and human health.

Thanks for the suggestion, done.

  • The turtle records are discussed in relation to the regions where they were reported. The study area should be described as such (i.e., 20 regions in Italy) and provide a study area map showing the regions. Perhaps separate this into 2 separate figures: (1) a study area map showing native distribution and translocation to Italian regions and (2) maps showing the distribution of each species. For the latter, can the symbols be enhanced (bold outline) as the green squares are difficult to see on the base map.

Thank you for your suggestions. However, we consider it appropriate to generally describe the study area (i.e., Italy), going into specifics by area (e.g., North, Centre and South) would be too long considering the type of manuscript, i.e., Communication.

The map has been modified with higher resolution. We believe that the current maps may be quite comprehensive; but we will keep his advice in mind for a subsequent, more comprehensive study (i.e., article).

  • As mentioned for Macrochelys in the subsequent paragraph, how many of these non-native Chelydra were removed from the wild? Is this common practice or is there some sort of protocol when capturing these non-native species?

As reported, these are reports in newspaper articles. In this regard, our intention is to map the distribution/recovery of these species in Italy (for the reasons mentioned above). Yes, there are collection protocols devised by the institutes, etc. involved, and the captured specimens entrusted to specialised centres (e.g., zoo, aquariums, etc.).

  • References to components of Figures 1 and 2 seem out of order (i.e., refer to Fig.1a and then Fig. 1b; refer to Fig 2a,b and then Fig. 2c,d). Either switch paragraphs (Macrochelys first, Chelydra second) or switch maps in Fig. 1 and photos in Fig. 2.

  • Thanks for the suggestion, done.

  • Table 1 lists 4 reports and there are 4 symbols represented on the map in Fig. 1a. It may be worth mentioning the turtles were captured and removed from the wild.

Thanks for the suggestion, done. Also, only 4 points are given because they represent the area of discovery, i.e., Lucinasco = 4 = 1 geographical point.

  • This doesn't make any sense, carapace lengths are expressed as meters or centimeters. Is one column length and the other weight? After accessing ref #57 it appears that the first column is weight (kg) and the second column is length (cm). Change descriptions accordingly.

Thanks for the suggestion, done. Yes it is a typo, we meant to write: # TW = total weight, TCL = total carapace length; expressed in kilograms (kg) and (cm), respectively.

  • This paragraph (lines 143-147) seems out of place and already mentioned in the introduction (?).

Also worth noting, the 2 recognized species of Macrochelys are proposed for listing as "threatened" in the U.S. Endangered Species Act:

Thanks for the suggestion, done.

  • Identified as mainly carnivorous in the Introduction. Also, ref #30 suggested vegetation could be ingested incidental to capture of animal prey.

Thanks for the suggestion, done.

  • Lines 156-188, this is a rather long paragraph and focuses on impacts to pet owners. Suggest reducing the length of this paragraph and emphasize the subsequent paragraph (lines 189-200) which provides better information on the possible health effects of pet turtles released into the wild.

Thanks for the suggestion, we will keep it in mind for the next study. We think that as a communication, it may be ok because it provides general information on possible health problems.

  • The following recent paper documents Chalmydia for T. scripta in Italy with reference to other pathogens spread by this turtle species: https://www.mdpi.com/2076-2615/12/6/784

Thanks for the suggestion, done.
